# Glycoprotein Profile Measured by a ^1^H-Nuclear Magnetic Resonance Based on Approach in Patients with Diabetes: A New Robust Method to Assess Inflammation

**DOI:** 10.3390/life11121407

**Published:** 2021-12-16

**Authors:** Núria Amigó, Rocío Fuertes-Martín, Ana Irene Malo, Núria Plana, Daiana Ibarretxe, Josefa Girona, Xavier Correig, Lluís Masana

**Affiliations:** 1Biosfer Teslab, Plaça del Prim 10, 2on 5a, 43201 Reus, Spain; rociofrtsm@gmail.com; 2Department of Basic Medical Sciences, Universitat Rovira i Virgili (URV), Institut d’Investigació Sanitària Pere Virgili (IISPV), Av. Universitat 1, 43204 Reus, Spain; 3Centro de Investigación Biomédica en Red de Diabetes y Enfermedades Metabólicas Asociadas (CIBERDEM), Instituto de Salud Carlos III (ISCIII), 28029 Madrid, Spain; anairenemalo@gmail.com (A.I.M.); nuria.plana@salutsantjoan.cat (N.P.); daiana.ibarretxe@urv.cat (D.I.); josefa.girona@urv.cat (J.G.); xavier.correig@urv.cat (X.C.); luis.masana@urv.cat (L.M.); 4Vascular Medicine and Metabolism Unit, Research Unit on Lipids and Atherosclerosis, Sant Joan University Hospital, Universitat Rovira i Virgili (URV), Institut d’Investigació Sanitària Pere Virgili (IISPV), 43201 Reus, Spain; 5Metabolomics Platform, Department of Electronic Engineering, Universitat Rovira i Virgili (URV), Institut d’Investigació Sanitària Pere Virgili (IISPV), 43007 Tarragona, Spain

**Keywords:** H-NMR, metabolomics, glycoproteins, diabetes, CRP

## Abstract

Patients with type 2 diabetes mellitus (T2DM) and atherogenic dyslipidemia (AD) are at higher risk of developing cardiovascular diseases (CVDs), so an interest in discovering inflammation biomarkers as indicators of processes related to CVD progression is increasing. This study aims (a) to characterize the plasma glycoprotein profile of a cohort of 504 participants, including patients with and without T2DM and/or AD and controls, and (b) to study the associations between the glycoprotein profile and other lipid and clinical variables in these populations. We characterized the plasma glycoprotein profiles by using ^1^H-NMR. We quantified the two peaks associated with the concentration of plasma glycoproteins (GlycA and GlycB) and their height/width ratios (H/W GlycA and H/W GlycB), as higher and narrower signals have been related to inflammation. We also quantified GlycF, the signal of which is proportional to the concentration of the acetyl groups of free N-acetylglucosamine, N-acetylgalactosamine, and N-acetylneuraminic in the samples. The lipoprotein profile was also determined (Liposcale^®^). Standard clinical and anthropometric measurements were taken. Multivariate classification models were developed to study the differences between the study groups. Reduced HDL-C levels, increased small dense LDL and HDL particles, and elevated TG levels were significantly associated with glycoprotein variables. Glycoprotein values in the diagnostic groups were significantly different from those in the CT groups. AD and DM conditions together contribute to a positive and significant synergetic effect on the GlycA area (<0.05) and the H/W ratios of GlycA (<0.01) and GlycB (<0.05). By adding the new glycoprotein variables to the traditionally used marker of inflammation C-reactive protein (CRP), the AUC increased sharply for classification models between the CT group and the rest (0.68 to 0.84), patients with and without dyslipidemia (0.54 to 0.86), and between patients with and without diabetes (0.55 to 0.75). ^1^H-NMR-derived glycoproteins can be used as possible markers of the degree of inflammation associated with T2DM and AD.

## 1. Introduction

Type 2 diabetes (T2DM) is one of the most common diseases and has been growing exponentially in recent years [1]. Patients with diabetes also have dyslipidemia, with an increased risk of cardiovascular diseases (CVD) [2,3]. The most common pattern of dyslipidemia in T2DM patients is atherogenic dyslipidemia (AD), which is characterized by elevated total triglycerides (TG) and small LDL particles, and decreased HDL cholesterol levels [4,5] with insulin resistance (IR) being the main pathophysiological mechanism [3]. On the other hand, T2DM is considered a chronic subclinical inflammatory state that also contributes to vascular complications [6,7].

C-reactive protein (CRP) can help to refine the global risk assessment for coronary heart disease (CHD), particularly among persons who are at intermediate risk on the basis of traditional risk factors alone. However, it has also been shown to be prone to fluctuations and is an inconsistent marker for predicting CVD in patients with chronic inflammatory diseases at an individual level [8]. For some years now, there has been increasing interest in inflammation biomarkers, which play a major role in the onset and progression of CVD [9]. Several studies have shown that glycoproteins play a key role in inflammatory and pathological processes, such as the genesis and/or progression of CVDs, and are also related to numerous pathologies, in particular cardiovascular risk factors, such as diabetes mellitus [10,11,12,13]. It has been reported that the signal in the 1H-NMR spectrum of glycated proteins is produced by the -COCH_3_ acetyl groups of N-acetylglucosamine and N-acetylgalactosamine (GlycA) and N-acetylneuraminic acid (GlycB) [14,15]. Unlike common biomarkers of inflammation, such as CRP or inflammatory cytokines, GlycA is a composite biomarker that integrates the protein levels and glycation states of several of the most abundant acute-phase proteins in serum (α1-acid glycoprotein, haptoglobin, α1-antitrypsin, α1-antichymotrypsin, and transferrin) [14,16]. This provides a more stable measurement of systemic inflammation with less intra-individual variability for GlycA than for high sensitivity CRP [14]. Although GlycA, and to a lesser extent GlycB, is the variable that has most been studied in terms of glycoproteins determined by 1H-NMR based on the metabolomic approach, other new variables have recently been reported, such as the derived parameters H/W GlycA and H/W GlycB, which are related to the shape of the signal (higher and narrower signals have been related to inflammation) [15].

Particularly, the scientific interest and number of studies demonstrating the strength of the NMR glycosylation profile as a biomarker for systemic inflammation has exponentially grown since the last decade [17]. Glycosylation has been associated with the increased risk of cardiometabolic diseases [18], cancer, and the worst evolution of non-communicable [19,20,21] and communicable diseases [22], even after its adjustment for other inflammatory markers and in pathological processes and biological conditions that are different in nature, such as immune alterations [23] and functional hyperandrogenism or obesity [24], which share the common thread of low grade inflammation.

In this study, we characterized the plasma 1H-NMR glycoprotein profile of T2DM patients with and without AD; we explored its association with their advanced NMR lipoprotein profile; and, finally, we compared the ability of these new NMR based on inflammatory markers to discriminate specific patterns between study groups, other than those commonly used in clinical settings, including traditional risk factors and CRP.

## 2. Results

Table 1 shows the demographic, clinical, and biochemical characteristics of the study population. Significant differences were observed between the groups in terms of age, and between the CT and the other groups in terms of body mass index (BMI), CRP, and Hb1AC. Lipid variables were generally higher in the study groups than in the CT group, except for HDL-C, which was higher in the CT group.

### 2.1. Associations of ^1^H-NMR-Derived Glycoprotein Variables with the Clinical and the Liposcale^®^ Test Variables

Table 2 shows the association between the five ^1^H-NMR glycoprotein variables and continuous clinical variables, such as BMI, blood glucose, CRP, HbA1c, and Liposcale^®^ (Biosfer Teslab, Reus, Spain). A similar pattern was observed in all five variables. We found no significant association between age and the concentration of glycated proteins in plasma. However, there was a positive and significant association between the concentration of glycated proteins and BMI, CRP, and blood glucose. For HbA1c, the trend was similar but only significant with the H/W GlycA (*p* < 0.05) and H/W GlycB (*p* < 0.01) ratios. Regarding the lipoprotein related parameters, we found that total abundance of GlycA, GlycB, and GlycF, and the shape of GlycA and GlycB were clearly associated with a proatherogenic pattern, which included a negative and significant association of the five glycoprotein parameters and HDL cholesterol (HDL-C) levels. The same applied to the medium HDL particle concentration and to the size of VLDL, HDL, and LDL particles.

Some of the independent variables considered in the three linear models significantly affected each glycoprotein variable. In model A, all the glycoprotein variables were significantly associated with BMI. Moreover, both the area and H/W ratio of GlycB were associated with age. In model B, all the glycoprotein variables were equally significantly and positively associated with BMI and diabetes. In model C, after the adjustment by lipid variables, we observed that all the glycoprotein variables were significantly and negatively associated with HDL-C, and significantly and positively associated with TG. However, some differences were observed between the glycoproteins in this model: the area and H/W ratio of GlycB were also significantly associated with gender, BMI, and diabetes; the area and H/W ratio of GlycF were significantly associated with gender, diabetes, LDL-C, and TG; the area of GlycA was only significantly associated with the three lipid variables and diabetes; and the GlycA H/W ratio with gender and BMI. (See Appendix A).

### 2.2. Analysis of Glycoproteins and CRP in the Study Population Groups

Significant differences were identified in the glycoprotein variables and CRP between the study groups (see Figure 1). The first thing to note is the difference in the dispersion of the variables: while the glycoprotein variables showed a narrow distribution, CRP clearly presented a wider dispersion. All glycoprotein variables showed similar and significant differences between the diagnostic groups, they were stronger when the AD status was compared (DM (−) AD (+)) and (DM (+) AD (−)); however, the CRP levels were not significantly different among the DM or DA groups. Neither the glycoprotein variable nor the CRP were significantly different between the groups (DM (+) AD (+)) and (DM (−) AD (+)).

All the glycoprotein parameters distinguished between having AD or not in a population with DM, and were also capable of detecting significant differences between the group that only had one of the conditions: DM or AD. These data indicated that the dominant condition was AD, and that the glycoprotein variables were much more affected when both conditions coexisted.

To examine the influence of AD and DM on each glycoprotein variable and CRP, we analyzed the main effect of each independent condition and its interaction. Appendix A shows the two-way Anova results for AD, DM, and the interaction between them for each variable. AD was significantly influenced by increasing the value of all the glycoprotein and CRP variables. The DM significantly affected the value of Area GlycB, H/W GlycB, H/W GlycA, and CRP. However, the interaction between both effects (AD and DM) was only significant for Area GlycA (*p* < 0.05), H/W GlycA (*p* < 0.01), and H/W GlycB (*p* < 0.05), which showed a synergistic effect between the two conditions.

To refine the separation between the groups, three different PLSDA models were built (see Figure 2) with the glycoprotein variables and CRP across the different clinical categories: model 1 (panel A) shows the CT versus the rest of the study population; model 2 (panel B) shows the individuals with DM versus individuals without DM; and model 3 (panel C) classifies the individuals with or without DA. 

For each of these models, we represented firstly the PLS-DA scores biplot in the space spanned by the two latent variables (LV1 vs. LV2), which reflected the classification of the samples. The LV1 explained more than 70% of the variance in the 3 models, while the LV2 explained between 9 and 13 of variance depending on the model. The biplot for model 1 (Figure 2, panel A) showed a clear separation between the CT and the rest of the study population. Model 2 (Figure 2, panel B) showed a good separation between the patients with and without AD if glycoprotein variables were considered, and a less clear separation if only CRP was considered. Model 3 (Figure 2, panel C) showed a weaker separation between the patients with and without DM. 

Secondly, we also presented the ROC curve plot (sensitivity versus 1-specificity) for each model, illustrating the diagnostic ability of the inflammatory variables between the two groups beyond the traditional risk factors. Six ROC curves were evaluated: one for CRP alone and the other five curves for CRP plus one or more glycoprotein. 

In all cases, the ^1^H-NMR glycoprotein variables greatly improved the separation between the groups. The AUC of the CRP ROC curve in the three models was much smaller than the AUC, when the glycoprotein variables were added (0.68 vs. 0.84 for model 1; 0.54 vs. 0.86 for model 2; and 0.55 vs. 0.75 for model 3).

## 3. Discussion

The enhanced synthesis of proinflammatory cytokines and acute-phase proteins characterizes the early stages of T2DM and increases as the disease progresses [6]. Many studies have suggested that the intra-individual reliability and variability of CRP is an inflammatory marker in the prediction and follow-up of some diseases [16,25]. In this study, we have characterized the ^1^H-NMR plasma glycoproteins of T2DM patients with and without AD, we have explored their relation with lipoproteins and clinical variables, and we have evaluated them as new emerging inflammatory markers of disease in comparison to CRP.

Our study showed that the glycoprotein profile was positively associated with being female, BMI, blood glucose, CRP, and a proatherogenic lipoprotein pattern with increased levels of LDL-C and TG. These results are in line with those of some studies in which GlycA has been associated with CRP [26] and a high BMI [27,28,29,30]. The association with HbA1c, however, was not so clear. There appeared to be a significant association only with the ratios H/W of GlycA and GlycB. However, HbA1c was significantly higher in the three study groups than in the control group, and even higher in the DM+ groups. These results suggest that the inflammatory pathways for HbA1c and serum/plasma glycoproteins are different and determined by ^1^H-NMR. More studies are needed to establish the bases of the inflammatory pathways of these blood proteins.

The significant negative associations found between the five glycoprotein variables and HDL-C and the size of VLDL, HDL, and LDL particles, and the high positive correlation with TG are in line with the literature [31,32], where diabetes is related to reduced HDL-C levels, a predominance of small dense LDL particles, and elevated TG levels [33]. In addition, this study found a significant positive association between glycoproteins and small HDL, the most abundant subclass in patients with T2DM [34]. These results suggest that an increase in glycosylated proteins is consistent with the characteristic lipid pattern of both T2DM and AD. In addition, linear regression models showed that both the area and H/W ratio of GlycB was largely associated with high TG concentrations and low HDL-C concentrations, while the area and H/W ratio of GlycA were also associated with high concentrations of LDL-C.

The significant differences in the glycoprotein values of each of the diagnostic groups versus the CT group were expected. However, the CRP showed greater dispersion among individuals from each group than the ^1^H-NMR glycoprotein variables, which confirms the higher intra-variability mentioned in the literature [16,25]. It appears that AD contributes to an increase in the five glycoprotein variables, while DM contributes to an increase in the GlycB area and the two variable H/W ratios, but not in the GlycA area. The GlycA area was associated with AD but not with DM. The two conditions together contribute to a positive and significant synergic effect on the GlycA area and the H/W ratios of GlycA and GlycB.

PLSDA models suggest that using the new glycoprotein variables in conjunction with common inflammation markers, such as CRP, maximizes the sensitivity and specificity of classifying DM and AD. This set of new inflammatory markers can be used to develop effective and novel strategies for disease intervention.

These results are in line with those of several studies in the literature, which report an increased concentration of GlycA in patients with T2DM [35,36]. The mechanisms by which these glycoprotein parameters increased in diabetes are not well understood. The α1-glycoprotein, one of the circulating glycoproteins detected by ^1^H-NMR, is known to be a marker glycemia because prolonged inflammation decreases glucose tolerance [37,38]. The close relationship of insulin sensitivity with GlycA, but not with GlycB, has also been described [39].

Nevertheless, our study is not free from certain limitations. The sample size and the distribution of the main anthropometrical cofounding factors, including age, gender, and BMI were not homogeneous between the groups. The univariant analysis was intrinsically affected by this asymmetry among groups, which directly represents what is generally found in clinical assessments for these specific patients, such as age, gender, and BMI, the main drivers for this metabolic syndrome related conditions. However, the regression models were properly adjusted in an attempt to minimize this variability. In addition, we have not been able to provide data on other clinical complications of diabetes because this was beyond the scope of this article. The concordance between alternative technologies to simultaneously measure glycosylated proteins, independently of the protein and sugar structure, is also limited because of the unique nature of the NMR. Among the strengths of this study, are the quality and novelty of the analytical procedures used and, from a single experiment with ^1^H-NMR, we can obtain both the glycoprotein variables and the advanced lipid profile, which leads to lower costs and less experimental variability. 

The present study compares protein glycosylation markers in relation to CRP, for assessing inflammation in specific metabolic conditions, opening the door to a future clinical application, although alternative inflammatory markers could potentially be implemented earlier in clinical settings. Therefore, these novel NMR glycoprotein parameters should be further compared in terms of improving inflammatory evaluation.

In conclusion, the present study suggests that ^1^H-NMR glycoproteins can be used as markers of the degree of inflammation associated with diabetes and dyslipidemia. Furthermore, we have shown that these new markers have a lower inter-individual variability; therefore, they can improve the predictive outcomes of CRP in patients with and without diabetes and in patients with and without dyslipidemia. Future studies will be necessary to understand the mechanisms underlying inflammation in AD and diabetes.

## 4. Materials and Methods

**Design and study subject:** For this cross-sectional study, we recruited 504 individuals attending the vascular medicine and metabolism unit of the University Hospital Sant Joan de Reus, for disorders associated with the lipid metabolism. Type 2 diabetes was diagnosed using standard clinical criteria. Atherogenic dyslipidemia was defined as TG > 150 mg/dL and HDL-c < 40 or 50 mg/dL for males and females, respectively. Subjects with chronic lung, renal or liver disease, cancer, or any other serious disease were excluded. Patients on lipid-lowering drugs underwent a 6 week wash-out period (8 weeks if they were on fibrates). Anamnesis, anthropometric, and physical examination data were recorded.

The entire population was classified into 4 groups, depending on whether they had T2DM (DM) and/or AD; DM and AD (DM (+) AD (+), n = 129); AD (DM (−) AD (+), n = 38); or DM (DM (+) AD (−), n = 222). A total of 115 patients had neither T2DM nor AD, and they were referred to as the control population (CT, n = 115). The study was approved by the randomized controlled trial and Clinical Investigation Committee of the Pere Virgili Institute for Health Research (IISPV), and all the participants signed the written consent form. 

**Clinical and standard biochemical variables:** Participants had their clinical history and their anthropometric data recorded and were subject to a physical examination. A blood sample was obtained in a fasting state and deeply frozen plasma and serum aliquots were stored in our research institute’s biobank until use. Standard biochemical determinations were performed using standard biochemical methods. If the patients were on lipid lowering therapy, the blood sample was obtained after a six week washout period.

**Glycoprotein and lipoprotein profiling by ^1^H-NMR:** For glycoprotein profiling, plasma samples were analyzed by ^1^H-NMR. Serum samples (200 μL) were previously diluted with 50 µL deuterated water and 300 µL of 50 mM phosphate buffer solution (PBS), at pH 7.4 consisting of 30.70 Na_2_HPO_4_ mM and 19.30 NaH_2_PO_4_ mM, before NMR analysis. The ^1^H-NMR spectra were recorded at 310 K on a Bruker Avance III 600 (Bruker BioSpin GmbH, Rheinstetten, Germany)spectrometer operating at a proton frequency of 600.20 MHz (14.1 T). One-dimensional ^1^H NMR pulse experiments were carried out, including nuclear Overhauser effect spectroscopy (NOESY) and longitudinal eddy-current delay (LED) pulses. To measure the glycoprotein variables, we analyzed the glycoprotein region of the ^1^H-NMR spectra, which is between 2.19 and 1.90 ppm, and represented the resonance of the N-acetyl methyl groups of the N-acetylglucosamine, N-acetylgalactosamine, and N-acetylneuraminic acid moieties on the carbohydrate portions of circulating glycoproteins, such as α1-acid glycoprotein, haptoglobin, α1-antitrypsin, α1-antichymotrypsin, and transferrin [14,15,40]. We measured the previously reported [15] glycoprotein variables, including areas of GlycB (Area GlycB), GlycA (Area GlycA), and the height/width ratios of GlycB and GlycA. The areas of GlycA and GlycB provided the acetyl group concentrations of protein-bound N-acetylglucosamine, N-acetylgalactosamine, and N-acetylneuraminic acid in plasma. In each case, the form of the function (H/W ratio) depended on the height, which was related to the concentration, and the width, which was related to the flexibility and the aggregation of the molecules generating the signal. This study describes a complementary new analytical function called GlycF, the area of which reflects the acetyl groups of free N-acetylglucosamine, N-acetylgalactosamine, and N-acetylneuraminic acid in the sample (that is to say, they do not bind to proteins).

We also obtained the lipoprotein profile using the NMR test Liposcale^®^ (CE), as previously reported [41]. This protocol evaluates the lipid concentrations (i.e., triglycerides and cholesterol), size and particle number of three different classes of lipoproteins (VLDL, LDL, and HDL), as well as the particle number of nine subclasses (large, medium, and small VLDL, LDL, and HDL). (Access to the NMR spectra: https://drive.google.com/drive/folders/1wfsryp6KbgWDnd-WM-6udjbHQzFMlLXj?usp=sharing, accessed on 15 December 2021).

**Statistical methods:** Firstly, we used the one-factor Anova with Bonferroni post-hoc correction or the χ2 test, depending on whether the variable was continuous or categorical, in order to study the distribution of the clinical variables. Spearman correlation coefficients were used to evaluate relationships involving the glycoprotein variables, the clinical variables, and the lipoprotein-related variables. A logarithmic transformation and an outlier test were performed for all those dependent variables that, according to the Lilliefors test, did not follow a normal distribution.

Secondly, a Wilcoxon–Mann–Whitney test was used to capture the differences in the glycoprotein profile and CRP values of each of the four groups. A Two-way Anova was performed, to examine the influence of the AD and DM conditions of each glycoprotein variable. 

**Multivariate data analysis:** In the first place, we used three linear regression models adjusted for the anthropometric and traditional risk factors of age, gender, BMI, DM, and traditional lipids to study the contribution of the clinical variables to each glycoprotein parameter. The independent variables considered were age, gender, and BMI for model A (i); model A + diabetes for model B (ii); and model A + LDL-C, HDL-C, and total triglycerides (TG) for model C (iii). The dependent variable (X) was each of the 5 ^1^H-NMR glycoprotein variables for each of the three models. The beta coefficients (β) determined the explanatory variable that had most weight in the explanation of each glycoprotein variable.

Partial least squares discriminant analysis (PLSDA) models were used as a supervised classification method between the study groups. PLS-DA relates the ***X*** matrix (experimental data) and the ***Y*** matrix (classes of samples), to find the maximum discrimination between classes (groups of study) and the maximum covariance between the ***X*** and ***Y*** matrices, simultaneously. [42] The area under the curve (AUC) was used to evaluate the capacity of the glycoprotein and CRP variables to distinguish between the two groups, beyond the anthropometric and traditional risk factors of age, BMI, and gender. All the models were auto scaled and cross-validated by the Venetian blinds cross validation method.

IBM SPSS^®^ (The Mathworks, Inc., Natick, MA, USA) statistics ver. 22 (IBM Corp. Armonk, NY, USA) was used to study the clinical statistical distribution between the groups and the adjustment of the linear regression models. Multivariate data analyses were computed in MATLAB (The Mathworks, Inc., Natick, MA, USA), ver. 7.10.0, using PLS-Toolbox, ver. 5.2.2 (Eigenvector Research, Inc., Wenatchee, WA, USA).

## Figures and Tables

**Figure 1 life-11-01407-f001:**
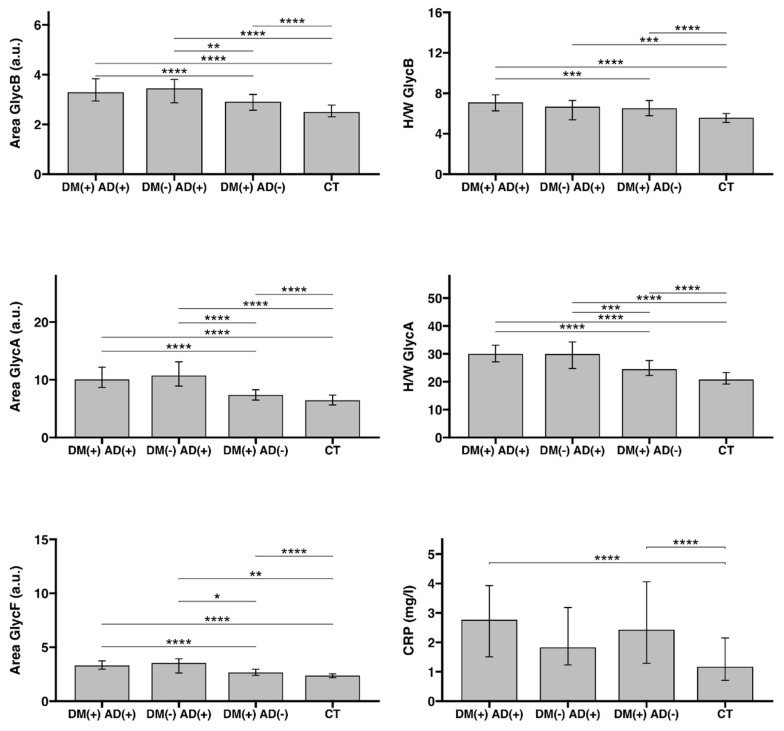
Univariant analysis of ^1^H-NMR glycoprotein variables and CRP between groups. The median and interquartile range (Iqr) are represented. The Wilcoxon–Mann–Whitney test has been used to calculate the significance (* *p* < 0.05, ** *p* < 0.01, *** *p* < 0.001, **** *p* < 0.0001).

**Figure 2 life-11-01407-f002:**
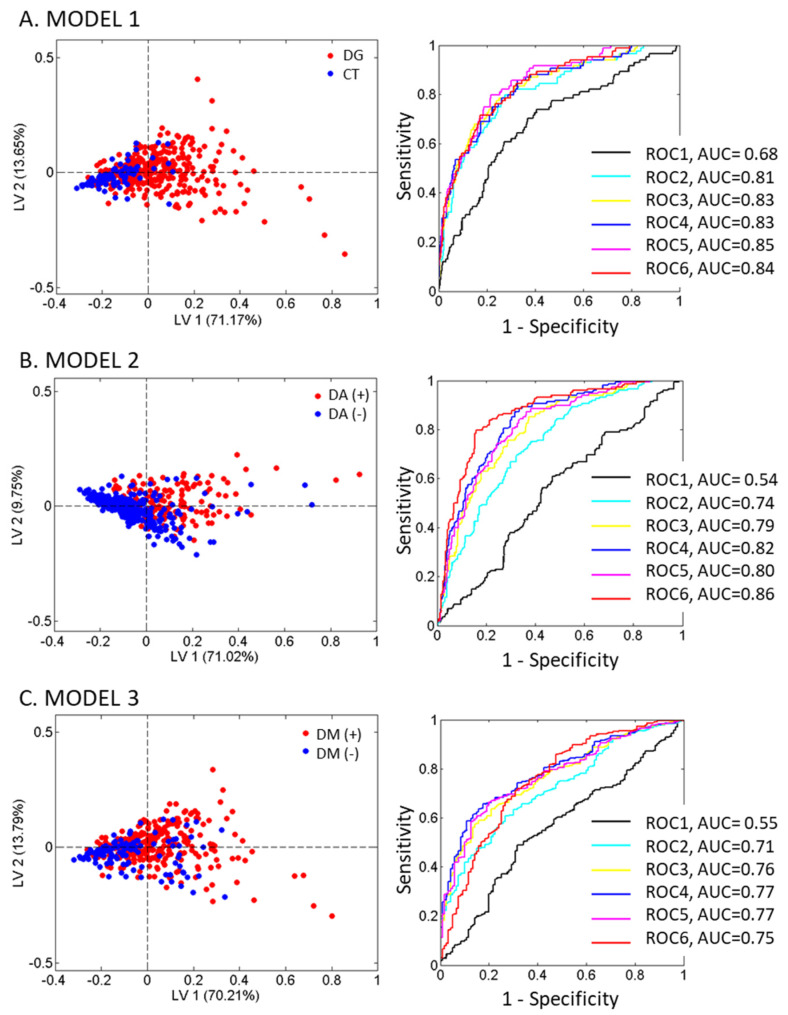
Abbreviations: DG, diagnostic groups; CT, control groups; AD, atherogenic dyslipidemia; and DM, diabetes mellitus. The biplots and ROC curves of the PLSDA models for matrix X with the traditional risk factors of age, BMI, and gender for all models, and the 5 1H-NMR glycoprotein variables and CRP. (ROC1: CRP; ROC2: CRP + Area GlycB; ROC3: CRP + Area GlycB + Area GlycA; ROC4: CRP + Area GlycB + Area GlycF + Area GlycA; ROC5: CRP + Area GlycB + Area GlycF + Area GlycA + H/W GlycB; and ROC6: CRP + Area GlycB + Area GlycF + Area GlycA + H/W GlycB + H/W GlycA).

**Table 1 life-11-01407-t001:** Distribution and clinical characteristic of groups.

	AD (+) DM (+) n = 129	DM (−) AD (+) n = 38	DM (+) AD (−) n = 222	CT n = 115	*p*
**Age (years)**	58 ± 14	54 ± 15	65 ± 10.75	58 ± 13.50	0.00 ** ^(A-B, A-D, B-C, C-D)^
**Gender (% male)**	50	33.3	49.5	61.4	0.02 *
**BMI (kg/m)**	31.85 ± 5.87	31.76 ± 2.86	30.20 ± 6.22	26.90 ± 4.49	0.00 ** ^(A-D, B-D, C-D)^
**Smoking (%)**	25.3	25	18	4.7	0.00 **
**Total-C**	215.37 ± 70.81	223.84 ± 47.54	197.19 ± 66.92	206.38 ± 43	0.00 ** ^(A-C, B-C, C-D)^
**LDL-C**	117.89 ± 59.25	125.14 ± 46.16	111.57 ± 46.27	107.79 ± 46.14	>0.05
**HDL-C**	41.04 ± 15.75	40.63 ± 16.53	51.54 ± 18.84	56.16 ± 22.93	0.00 ** ^(A-C, A-D, B-C, B-D)^
**TG**	215.56 ±127.62	217.10 ± 134.97	111.35 ± 57	71.48 ± 42.84	0.00 ** ^(A-C, A-D, B-C)^
**Glucose**	158.50 ± 57	114 ± 22	151.50 ± 62.25	101.50 ± 18	0.00 ** ^(A-B, A-D, B-C, C-D)^
**Hb1AC (%)**	6.70 ± 1.50	5.20 ± 0.60	6.90 ± 1.67	5.23 ± 0.31	0.00 ** ^(A-B, A-D, B-C, B-D, C-D)^

^1^ Abbreviations: BMI, body mass index; Hb1AC, glycated hemoglobin; A, AD(+)DM(+); B, AD(+)DM(−); C, AD(−)DM(+); and D, CT. (A) DM (+) DA (+), (B) DM (−) AD (+), (C) DM (+) AD (−), (D) CT. The median and the interquartile range (±) are indicated for continuous variables and the distribution percentage is indicated for categorical variables. *p*-values are corrected by the Bonferroni method. For categorical variables, the chi-squared *p*-value is shown. * is marked for *p* < 0.05 and ** for *p* < 0.01.

**Table 2 life-11-01407-t002:** Relationship between the five ^1^H-NMR glycoprotein variables and continuous clinical variables.

	AREA GLYCB		AREA GLYCF		AREA GLYCA		H/W GLYCB		H/W GLYCA	
	r	*p*	r	*p*	r	*p*	r	*p*	r	*p*
**CLINICAL VARIABLES**										
AGE	−0.03	0.46	−0.01	0.84	−0.05	0.31	0.02	0.75	−0.02	0.66
BMI	0.30	0.00 **	0.30	0.00 **	0.34	0.00 **	0.29	0.00 **	0.32	0.00 **
GLUCOSE	0.38	0.00 **	0.44	0.00 **	0.42	0.00 **	0.39	0.00 **	0.44	0.00 **
CRP	0.37	0.00 **	0.32	0.00 **	0.31	0.00 **	0.42	0.00 **	0.39	0.00 **
HBA1C	0.10	0.17	0.15	0.05	0.09	0.25	0.28	0.00 **	0.16	0.04 *
**LIPOSCALE^®^ VARIABLES**										
VLDL-C	0.61	0.00 **	0.76	0.00 **	0.88	0.00 **	0.42	0.00 **	0.74	0.00 **
IDL-C	0.49	0.00 **	0.48	0.00 **	0.55	0.00 **	0.31	0.00 **	0.49	0.00 **
LDL-C	0.04	0.30	0.08	0.04 *	0.13	0.00 **	−0.02	0.66	0.10	0.02 *
HDL-C	−0.40	0.00 **	−0.47	0.00 **	−0.56	0.00 **	−0.34	0.00 **	−0.49	0.00 **
VLDL-TG	0.60	0.00 **	0.79	0.00 **	0.89	0.00 **	0.44	0.00 **	0.75	0.00 **
IDL-TG	0.59	0.00 **	0.61	0.00 **	0.70	0.00 **	0.40	0.00 **	0.63	0.00 **
LDL-TG	0.41	0.00 **	0.35	0.00 **	0.42	0.00 **	0.25	0.00 **	0.39	0.00 **
HDL-TG	0.42	0.00 **	0.52	0.00 **	0.58	0.00 **	0.24	0.00 **	0.48	0.00 **
VLDL-P (nmol/L)	0.60	0.00 **	0.79	0.00 **	0.89	0.00 **	0.44	0.00 **	0.75	0.00 **
Large VLDL-P (nmol/L)	0.59	0.00 **	0.76	0.00 **	0.88	0.00 **	0.44	0.00 **	0.75	0.00 **
Medium VLDL-P (nmol/L)	0.61	0.00 **	0.78	0.00 **	0.90	0.00 **	0.44	0.00 **	0.75	0.00 **
Small VLDL-P (nmol/L)	0.60	0.00 **	0.79	0.00 **	0.89	0.00 **	0.44	0.00 **	0.74	0.00 **
LDL-P (nmol/L)	0.14	0.00 **	0.19	0.00 **	0.23	0.00 **	0.06	0.16	0.20	0.00 **
Large LDL-P (nmol/L)	0.03	0.53	−0.01	0.89	0.06	0.17	−0.06	0.12	0.03	0.54
Medium LDL-P (nmol/L)	0.01	0.88	0.03	0.47	0.07	0.10	−0.03	0.50	0.06	0.15
Small LDL-P (nmol/L)	0.24	0.00 **	0.31	0.00 **	0.35	0.00 **	0.14	0.00 **	0.31	0.00 **
HDL-P (μmol/L)	−0.09	0.03 *	−0.06	0.17	−0.09	0.03 *	−0.14	0.00 **	−0.10	0.02 *
Large HDL-P (μmol/L)	0.19	0.00 **	0.17	0.00 **	0.27	0.00 **	0.06	0.16	0.17	0.00 **
Medium HDL-P (μmol/L)	−0.36	0.00 **	−0.46	0.00 **	−0.55	0.00 **	−0.31	0.00 **	−0.48	0.00 **
Small HDL-P (μmol/L)	0.10	0.01 *	0.21	0.00 **	0.23	0.00 **	0.00	0.96	0.17	0.00 **
VLDL-Z (nm)	−0.27	0.00 **	−0.44	0.00 **	−0.40	0.00 **	−0.21	0.00 **	−0.33	0.00 **
LDL-Z (nm)	−0.38	0.00 **	−0.53	0.00 **	−0.51	0.00 **	−0.35	0.00 **	−0.47	0.00 **
HDL-Z (nm)	−0.45	0.00 **	−0.62	0.00 **	−0.74	0.00 **	−0.34	0.00 **	−0.61	0.00 **
NON-HDL-P (nmol/L)	0.29	0.00 **	0.37	0.00 **	0.43	0.00 **	0.17	0.00 **	0.38	0.00 **
TOTAL-P/HDL-P	0.28	0.00 **	0.32	0.00 **	0.39	0.00 **	0.21	0.00 **	0.35	0.00 **
LDL-P/HDL-P	0.17	0.00 **	0.19	0.00 **	0.25	0.00 **	0.13	0.00 **	0.23	0.00 **

^1^ Spearman correlation coefficients (r) and Spearman *p*-values (*p*) are shown. Significance is marked (* for *p* < 0.05 and ** for *p* < 0.01). Green indicates positive associations, while orange indicates opposite associations. For Age, IMC, and Glucose n = 455; for CRP n = 396; for HbA1c n = 172; and for the other variables n = 504.

## Data Availability

NMR analysed spectra are publicly available: https://drive.google.com/drive/folders/1wfsryp6KbgWDnd-WM-6udjbHQzFMlLXj?usp=sharing (accessed on 15 December 2021).

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
