# Peer review of "Glycoprotein Profile Measured by a 1H-Nuclear Magnetic Resonance Based on Approach in Patients with Diabetes: A New Robust Method to Assess Inflammation"

_life, 2021, doi:10.3390/life11121407_

Round 1

Reviewer 1 Report

In this study authors characterize the plasma glycoprotein profile of a cohort of 504 participants and studied the associations between the glycoprotein profile and other lipid and clinical variables. I have following comments:

(a) The 1D and 2D NMR Data (NOESY) is missing in the supplementary information. The authors should provide the data via stacking or as zip file.

(b) The authors should provide in the discussion, how their method is comparable to other analytical methods such as using LC-HRMS etc.

Reviewer 2 Report

In this manuscript, the authors used 1H-nuclear magnetic resonance based method to measure glycoprotein profile as a biomarker of inflammation. Although the approach itself is interesting, there are things that needs further evidence to support their hypothesis.

1) It is not immediately apparent how glycoprotein is related to inflammation. The authors only provide one or two sentence on this matter, but to support their hypothesis it is required to provide strong evidence on why glycoproteins are important for accessing inflammatory levels.

2) The authors say 'a new robust method to assess inflammation' but they only use CRP for the analysis. Although CRP is a marker frequently used for assessing inflammation, there are plenty of other inflammatory biomarkers which are important in progression of DM and AD. The authors shoudl at least adopt a few more to compare with CRP levels and to confirm that the glycoproteins are indeed useful.

3) The major limitation of this study is that the results are not adjusted by the confounding factors. The age, gender, smoking status is all significantly different among groups, so whether these factors are having an influence on the results are not resolved.

4) In addition to the above comment, the correlations between glycoprotein and CRP should be further analyzed normalizing for other parameters to show that CRP levels are the dominant marker related to glycoprotein levels, independent of other variables. The focus at this point seems to be more on the lipid profiles than inflammation.

5) Figure 1 should be presented in a way that the difference between the groups can be seen.

6) How do the authors explain the fact that neither glycoproteins nor CRP can distinguish between DM and AD? Would it not be a potential problem when applying in clinic?

Reviewer 3 Report

  1. The age, gender, and BMI values of the groups are incompatible. The authors should clarify whether this affects the parameters they studied. Additions should be made to the Discussion section.
  2. The mean glucose value of the AD (+) group was 114 mg/dl, was OGTT performed in this group? how was it excluded whether there was a diabetic patient or not?
  3. The Hb1AC value of the control group is too low.
  4. The English version of the article needs to be revised.

Round 2

Reviewer 2 Report

Although some additional experiments required were not performed, the authors have fully answered the reviewer's comments.

Reviewer 3 Report

I have no further suggestions